# AUTOREGRESSIVE KNOWLEDGE BASE COMPLETION

## ABSTRACT

Many Knowledge Graphs (KGs), despite their size, remain highly incomplete. This problem has motivated many approaches to *complete* the KGs by embedding their constituents in a latent space to find missing links. While these methods perform well on the commonly used metrics, they rank the missing links with scores that are uncalibrated and context-dependent. The fact that the scores are "local", in the sense that they relate to a specific context, makes it difficult to determine the final link truth value and to answer complex queries. Another limitation is that their learning depends on negative sampling, which is challenging due to the Open World Assumption (OWA).

To solve these issues, we propose a novel auto-regressive generative model that learns a joint distribution of the entities and relations of the KG without resorting to negative sampling. This distribution allows us to compute "global" scores for the missing links which are calibrated and interpretable in different contexts. Moreover, our method has the advantage that it offers probabilistic semantics for complex reasoning and knowledge base completion. Finally, our empirical evaluation shows that although our method is not suitable for KGs with a regular topology that can be easily learned by prior local methods, it excels with KGs that have a more complex structure with many inter-contextual dependencies, achieving or even surpassing state-of-the-art performance on KB completion and with consistent scores across the entire KG.

## 1 INTRODUCTION

**Motivation** Knowledge Graphs (KGs) are structured representations of knowledge organized as graphs which model relationships between entities. KGs are highly popular and widely used in various domains, from search engines and recommendation systems to data integration Hogan et al. (2021). Currently, KGs are widely studied in academia and extensively utilized in industry, driving advancements in both research and practical applications Zhang et al. (2023); Wang et al. (2017b).

**Problem** An important problem that limits the usefulness of KGs in real-world scenarios is that they remain highly incomplete, i.e., many links (and/or nodes) are missing. An important step to solve this problem consists of predicting, given a KG as input, which of the unknown links is likely to be true (Lü & Zhou, 2011). To this end, the mainstream approach consists of learning latent representations of the KG to *rank* effectively potential missing links. These methods, known as Knowledge Graph Embeddings (KGEs) (Wang et al., 2017a), transform the set of entities and relationships into continuous vector spaces, and then rank the missing links by evaluating their likelihood with a given scoring function. Conceptually, several prominent KGE approaches can be seen as binary classifiers since their primary objective is to construct models where each link corresponds to a distinct Bernoulli random variable (Trouillon et al., 2016; Nickel et al., 2011; Lacroix et al., 2018; Nguyen et al., 2018; Bordes et al., 2013c). For example, given a KG with a set of entities $\mathcal{E}$ and relations $\mathcal{R}$, a link from a source $s \in \mathcal{E}$ to destination $d \in \mathcal{E}$ with a specific relation type $r \in \mathcal{R}$, denoted as $\langle s, r, d \rangle$, is assigned to the variable $Y_{\langle s,r,d \rangle}$.

A major strength of KGE methods lies in their ability to use the likelihoods calculated by these models as reliable scoring mechanisms for ranking potential missing links. However, an important limitation is that they rely on the generation of false links for training the model, which is challenging to do under the Open World Assumption (OWA). Typically, these models resort to employing randomly generated links as false evidence, which is a technique known as negative sampling (Bor-

des et al., 2013b). An additional limitation is that these models lack a consistent *global* probabilistic interpretation of the scores (Friedman & den Broeck, 2020; Zhu et al., 2023), hence we say that their scores have a *local* interpretation. The fact that scores have only a local interpretation becomes particularly problematic when, for instance, we need to use the scores to determine the final truth value of the links by selecting an optimal threshold value or for answering complex queries since the scores in different domains cannot be compared(Arakelyan et al., 2023).

Both limitations have received considerable attention in the community. Some techniques have investigated the value of calibrating the scores in a post-processing phase to reduce the differences between domains Zhu et al. (2023); Arakelyan et al. (2023). Others have proposed to exploit the inherent structure of certain tensor factorization models (Lacroix et al., 2018; Nickel et al., 2011; Balažević et al., 2019; Trouillon et al., 2016) to normalize the scoring functions. By normalizing the scoring function, the edges are no longer modelled with separate binary variables but one random variable $Y$ (Loconte et al., 2023). In this last case, the idea is to learn a distribution $p(Y, S, R, D)$ (for modeling the truth value, source, relation type and destination of the link, respectively) and to consider $p(Y = true, S = s, R = r, D = d)$ as the scoring function for the link $\langle s, r, d \rangle$. Note that any joint distribution $p(Y, X)$ can be rewritten into $p(Y|X) \cdot p(X)$. Since the former is a classification component ($p(Y|X)$) while the latter is a generative one ($p(X)$) we refer to such an approach as an *hybrid* model Lasserre et al. (2006). While this approach yields global scores (since we have a single random variable $Y$), it still depends on negative sampling for training the classification component. Hybrid models learn the data distribution to improve the decision making process, however, learning the distribution of false edges is problematic if we only know which edges are true.

**Contributions** To overcome the limitations stated above, we propose a new generative method that learns directly the distribution $p(S, R, D|Y = 1)$ or more simply $p(S, R, D)$. We argue that this method has a benefit over the current ones since it produces *global* scores without resorting to negative samplings during learning. To learn the joint distribution, we first decompose it as $p(S, R, D) = p(S) \cdot p(R \mid S) \cdot p(D \mid R, S)$. Then, we employ (deep) AutoRegressive Models Gregor et al. (2014); Tomczak (2022) to learn the conditional distributions $p(R \mid S)$ and $p(D \mid R, S)$, while for the marginal distribution $p(S)$ we employ different strategies with the source entities in the training data set to learn a function that resembles the true marginal distribution.

We empirically demonstrate that our method yields significantly more consistent scores across different domains by assigning high probabilities only to links deemed highly likely to be true, as indicated by their top positions in local rankings. Additionally, our results achieve the best *global* performance in terms of Mean Average Precision. We consolidate unseen facts into a single global ranking and evaluate the truth value of the *top-k* positions in this ranking. Our method is competitive in link prediction (local rankings) using standard metrics; it reaches state-of-the-art performance when considering the consistency of the probabilities, making it particularly suitable for more complex processing tasks such as Knowledge Base Completion or Complex Query Answering.

## 2 Preliminaries

**Knowledge Graph Embedding Models** A common way to represent information in a KG is through triples. A triple $(s, r, d)$ consists of a source (subject), relation (predicate) and destination (object), where the sources and destinations come from the same set of entities, $s, d \in \mathcal{E}$ and $r \in \mathcal{R}$, the set of all relations. Every edge can be precisely characterized by a triple of the form $(s_i, r_i, d_i)$, and a KG is a subset of $\mathcal{E} \times \mathcal{R} \times \mathcal{E}$.

KGE models consist of two components. One component represents entities and relations as continuous vectors (embeddings) in a low-dimensional space. These embeddings are designed to capture the semantic meanings and structural information of the entities and relations while preserving the inherent structure of a KG. The second component is a scoring function that assigns a score to the vector representation of every triple, $\phi(\mathbf{e_s}, \mathbf{r_r}, \mathbf{e_d})$. That score is usually a real number and represents the model's belief in the truthfulness of the triple, typically in contrast to the score of other triples with the same source and relation.

**Probabilistic interpretation** The embeddings are in a continuous space. However, we can interpret them as categorical variables because they are direct mappings from the entities and relations.

Let $S$ be a random variable with values in $\mathcal{E}$, $R$ a random variable with values in $\mathcal{R}$, and $D$ a random variable with values in $\mathcal{E}$. The collection of all triples can be described by the joint distribution $p(S, R, D)$. According to the general product rule, we can factorize this distribution, for example, in $p(S) \cdot p(R|S) \cdot p(D|R, S)$ or $p(R) \cdot p(D|R) \cdot p(S|R, D)$, etc.

**Deep AutoRegressive Models** AutoRegressive models are sequence models that originate from time-series models Ho & Xie (1998), where observations from previous steps are used to predict the value at the current time. However, as elaborated in the notes by Grover (2018), they have also proven useful in modeling joint distributions when the data is not inherently sequential (van den Oord et al., 2016; Salimans et al., 2017). To do this, the model assumes that the conditional distributions correspond to a specific random variable, for instance, a Bernoulli random variable, which means that we limit the expressiveness of the conditional distribution. Then, we can approximate the distribution using parameterized functions e.g., neural networks, to represent the conditionals. To enforce that assumption, we can use a Softmax for categorical variables or a Sigmoid for Bernoulli random variables. The benefit of this assumption is that the entire joint distribution does not require global normalization. By ensuring that each conditional distribution is a valid and restricted distribution, the global normalization is enforced locally.

**Deep AutoRegressive Models for Knowledge Graphs** Although a KG only has three variables, specifying a probability for all possible triples of $\mathcal{E} \times \mathcal{R} \times \mathcal{R}$ quickly becomes infeasible, even for small-size KGs Loconte et al. (2023). Fortunately, the KG data is inherently categorical. Therefore, the assumption that random variables have to be Bernoulli is not an issue, and an autoregressive model is well suited to approximate the joint distribution $p(S, R, O)$ in one of the factorized forms.

**Optimization** Generative models optimize the distance between the data and the model distribution. By assuming that points in the dataset are sampled i.i.d. from the true data distribution, we obtain an unbiased Monte Carlo Maximum Likelihood Objective.

The goal of MLE objective is to find the parameters, $\theta$, that maximize the log-probability of the observed data points. Generative models are designed to generate new data points from the learned distribution. Generating new data points in AutoRegressive models is an iterative and, thus, slow process. Besides that we use an AutoRegressive model to learn only two variables i.e., two iterations, we do not require generating *new* triples. We are only interested in evaluating the likelihood of a triple, which can be done in parallel.

## 3 RELATED WORK

One of the motivations for implementing an AutoRegressive model using Convolutions, as discussed in Section 4, is to highlight its similarity to ConvE (Dettmers et al., 2018). Both approaches utilize Convolutions on embedding vectors. ConvE estimates the distribution $p(Y_{sr}|S, R)$, treating each edge as a Bernoulli random variable (Loconte et al., 2023). Our model replaces the Sigmoid function with a Softmax function to learn $p(D|S, R)$. Additionally, we introduce a conditional distribution $p(R|S)$ and incorporate an exogenous prior $p(S)$. Furthermore, we do not rely on generating artificial negative triples, our approach focuses on modeling the distribution of the triples directly, moving away from classification.

Other methods have applied AutoRegressive models to assign a likelihood to triples (Chen et al., 2021; Yao et al., 2019; Tresp et al., 2021; You et al., 2018). Those methods either use Large Language Models, model local subgraphs, or do not interpret the data describing a KG as a collection of three *categorical* variables. In this study, we focus explicitly on methods that learn a joint distribution of three *categorical* random variables and are trained using MLE. To the best of our knowledge, this has only been accomplished by Loconte et al. (2023).

They use a different class of generative models to model the data distribution: Energy-Based Models (EBMs). Contrary to AutoRegressive models, there is no restriction on the expressiveness of the function representing the distribution. Naturally, the function must still be a valid probability distri-

bution where the total outcome equals one and all probabilities are larger or equal to zero. EBMs, thus, are more expressive but require global normalization. Calculating the normalization constant for KGs, i.e. evaluating all possible triples, requires the summation of $\mathcal{E} \times \mathcal{R} \times \mathcal{E}$ triples at every training step and is infeasible even for small KGs. Exploiting the inherent structure of certain KGE models, namely viewing them as parameterized structured computational graphs circuits, they are able to calculate the normalization constant in $O((|\mathcal{E}|+|\mathcal{R}|) \cdot cost(\phi))$ time. With this normalization, they converted the binary classification models $p(Y|X)$ into generative models $p(Y, X)$.

The tensor factorization methods that they convert typically have simple scoring functions, e.g, the scoring function of `Complex` (Trouillon et al., 2016) is simply the (complex) dot product of the embedding vectors. This can be considered a feature when the goal is to learn a relatively simple ranking function. When learning a more intricate function, such as a joint distribution, this can become a limitation; by using neural networks to model the joint distribution, we can have more parameters in the scoring function to potentially better capture this function.

TractOR (Friedman & den Broeck, 2020) decomposes the binary classification models by assuming conditional independence, such that $p(Y_{sro} = 1|S, R, D) = p(E_s = 1|S) \cdot p(T_r = 1|R) \cdot p(E_o = 1|D)$. Although conditional independence is a strong assumption and TractOR is not a probabilistic model but a decomposition, they do meet the assumption of Probabilistic Databases that require every triple to be an independent Bernoulli random variable. Also, the decomposition into unary statements ensures fast computation of the probability of a complex query in Probabilistic Databases.

## 4 AUTOREGRESSIVE KNOWLEDGE GRAPH EMBEDDING MODELS

In this section, we describe the challenges of designing an AutoRegressive KGE model called `ART` and motivate our design decisions.

### 4.1 A DECOMPOSITION TRADE-OFF

A joint distribution $p(S, R, D)$ can be factorised into multiple equivalent decompositions. Two such decompositions are $p(R) \cdot p(S|R) \cdot p(D|R, S)$ and $p(S) \cdot p(R|S) \cdot p(D|R, S)$. Although equivalent, they have different implications for modelling the data that describes a KG. The conditional distributions can be *learned* with neural networks, while the marginal distribution can not. The marginal distribution can be modelled with a prior. A KG usually has significantly fewer relations than entities e.g., WikiData5M (Wang et al., 2021) has 5 Million entities and just 822 relations. Therefore, the number of facts for every $r$ is, on average, higher than the number of facts for every source. Ideally, we want to place the prior on $r$ such that the prior distribution reflects the true marginal; this because there is relatively more information on this variable, resulting in a more informative prior. Unfortunately, this comes at a great cost. Namely, this means that the complexity of modelling the second conditional distribution increases drastically. Because when modelling the second conditional as $p(S|R)$, we now have to produce an output for $|\mathcal{E}| * |\mathcal{E}|$ values instead of the $|\mathcal{R}| * |\mathcal{E}|$ outputs when modelling the second conditional as $p(R|S)$. Because typical real-world KGs are large, we put the prior on $s$. However, if the graph is small-scale, we advise placing the prior on $r$ for the best results.

### 4.2 PRIOR SELECTION

Due to the fact that the marginal distribution, $p(S)$, by definition, is not conditioned on another random variable, this component does not have embeddings as input. This does not preclude exploring several strategies to find a prior that accurately describes the true marginal distribution. The prior function can be any function based on background knowledge as long as it is informative of the true marginal $p(S)$. In this work, we opt for a frequentist approach and use the proportion of sources (subjects) in the train set as our prior for modelling $p(S)$. When working with infinite data and assuming that the distribution in the data is similar to that of the train set, this will result in a prior that resembles the true distribution. However, when we know or observe that the proportion is not an informative prior, we want to reduce the impact of the prior, $p(s)$, on the overall prediction $p(s) \cdot p(r|s) \cdot p(d|r, s)$. In its extreme, this is a uniform prior; every source is considered equally likely. A less extreme approach is to smoothen the function with a Softmax; the degree of smoothness can be controlled via a hyper-parameter $T$, often referred to as the Temperature (Renze & Guven, 2024). A more sophisticated approach is to model every entity with a logit (learnable

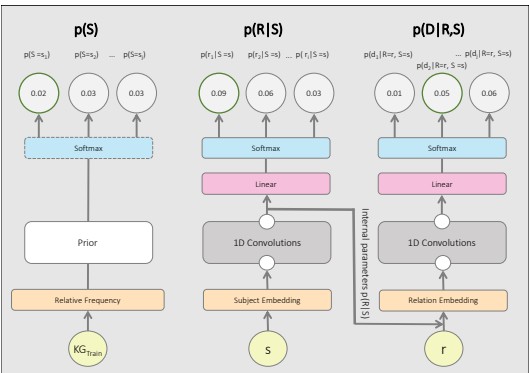

Figure 1: The architecture of ARC. The internal parameters used to model $p(R|S)$ are added as another input channel to $p(D|R, S)$. Because we model the data distribution, $p(X)$, there is only one correct label for all outputs, unlike binary classification methods, where multiple entities can be labelled true.

parameter) and apply a Softmax for a valid marginal distribution. Since no embeddings are used in the input, this will resemble the same frequency distribution introduced above. However, the prior is now internalized in the loss function and hence, the prior is no longer exogenous, but *learned* from data similar to the other two components, which can result in a more coherent joint distribution.

### 4.3 DIRECTIONALITY AND FAST INFERENCE

By inferring $p(s) \cdot p(r|s) \cdot p(d|r,s) \sim p(S, R, D)$, our model can assign a probability to any given triple. However, KGE methods typically require scoring all candidate entities given a specific entity and relation, as in the cases of head and tail prediction in link prediction tasks.

For tail prediction, our model allows straightforward computation by selecting the relevant variables and evaluating $p(S = s) \cdot p(R = r|S = s) \cdot p(D =?|R = r, S = s)$. Since the first two components are constant, and the third component is a Softmax over all entities, we can efficiently compute the joint probability of all candidates simultaneously.

In contrast, for head prediction $rank(d|r,s)$, the values for all three components differ, requiring the triples to be collected iteratively. While this approach is feasible and valid, it is computationally slow. To align with the standard KGE framework, we introduce an inverse triple for every KG, a common practice introduced in Dettmers et al. (2018). Notably, $rank(d|r,s)$ is equivalent to $rank(s|r^{-1}, d)$, allowing us to score all candidates simultaneously for head prediction, avoiding the need for iterative processing. This approach also adds directionality to our model, enabling different confidence levels for the same triple, which is sometimes useful Trouillon et al. (2016).

### 4.4 MODELLING THE CONDITIONAL DISTRIBUTIONS WITH NEURAL NETWORKS

The Neural Networks can be of arbitrary choice as long as they have shared parameterization, can process embeddings and are AutoRegressive. The shared parametrization (see fig. 1) is crucial to learning one joint distribution instead of two independent distributions. We choose a vanilla encoder-only Transformer (Vaswani et al., 2017) model, which we call ART, and implement a Convolutional model, which we call ARC, fullfilling these requirements. Figure 1 illustrates ARC. For the Transformer, we use one attention head, do not use positional encoding, and use only a small number of blocks. For the Convolutional model, we stack two 1D convolutions.

We arrive at the following learning objective:

$$\mathcal{L}_{\text{MLE}}(\theta) = \max_{\theta} \frac{1}{|\mathcal{G}|} \sum_{(s,r,d) \in \mathcal{G}} \log p_{\theta_0}(s) + p_{\theta_1}(r \mid s) + p_{\theta_2}(d \mid r, s) \tag{1}$$

Table 1: Statistics of the standard link prediction benchmarks `FB15k-237` (Toutanova & Chen, 2015) and `WN18-RR` (Dettmers et al., 2018) and `OGBLBioKG` (Hu et al., 2021)

| Dataset | Train-KG | Valid-KG | Test-KG | $\mathcal{E}$ | $\mathcal{R}$ |
|---------|----------|----------|---------|------|------|
| `FB15k-237` | 272,115 | 17,535 | 20,466 | 14,541 | 237 |
| `WN18-RR` | 86,838 | 3,034 | 3,134 | 40,943 | 11 |
| `OGBLBioKG` | 4M | 163K | 163K | 93,773 | 51 |

## 5 EMPIRICAL EVALUATION

Our goal is to assess the ability of our method to perform link prediction *globally*, and not locally (i.e., query-based) as done for prior methods, since it is precisely the former ability that allows us to support more effectively complex and inter-contextual query answer and to perform KB completion across the entire scope of the KB.

### 5.1 EXPERIMENTAL SETTING

**Datasets** We consider benchmark datasets that are commonly used in the literature for link prediction, which are `FB15K-237` Toutanova & Chen (2015), `WN18-RR` Dettmers et al. (2018), and `OGBLBioKG` Hu et al. (2021). See Table 1 for statistics. We follow the standard train-test split protocol, in which entities in the test set must appear in the training set, ensuring no new entities are introduced during evaluation. All metrics are computed under the *filtered* setting Bordes et al. (2013a). All experiments are run on a NVIDIA RTX A6000 GPU with 48GB of onboard memory.

**Baselines** We selected three baselines: `NBF` (Zhu et al., 2022), `ComplEx` (Trouillon et al., 2016), and `ComplEx`[2] (Loconte et al., 2023). `NBF` is chosen as it has reported the best results for (local) query-based link prediction. We include `ComplEx`, another state-of-the-art-method, due to its widespread usage in the literature and `ComplEx`[2], which is a variant of `CompleEx` with a probabilistic interpretation and which employs, like us, the Maximum Likelihood Estimation (MLE) as training objective. For the results on `ARC` please refer to Appendix B.

**Hyper-parameters** We fix the the batch size to $1024$ and the rank to $150$. We use AdamW (Loshchilov & Hutter, 2019) as optimizer and initialize the weights by sampling from a Dirichlet distribution Loconte et al. (2023). We randomly searched for the optimal learning rate and tried a small selection of model configurations. See Appendix C.1 for more details.

### 5.2 RESULTS ON GLOBAL LINK PREDICTION

**Task definition.** Prior art has been mostly evaluated on query-based local link prediction. Essentially, this task consists of ranking all possible completions for a given query and then assess the position in the ranking of known true triples. While this method is useful to evaluate the predictive power, it does not evaluate the ability of producing consistent scores across queries, which may be needed in a downstream task, e.g., for complex query answering or global KB completion.

To fix this problem, we evaluate our method differently. Instead of evaluating using many rankings, we construct *one* global ranking of facts by executing all queries from the test set and merging the obtained scores together. Like in traditional link prediction, we filter out the triples from the train and validation graph and duplicates (filtered settings). Moreover, we assume that unknown facts are false, such that we can borrow standard metrics from Information Retrieval to evaluate the ability of the models to rank the facts globally. This is an assumption adopted in the related literature as well. We use the mean-average-precision (MAP), and find the optimal threshold for which every fact is considered true or false. We optimise the precision and recall curve by considering as many thresholds as there are unique scores. We can then calculate the optimal $F1*$ score that balances precision and recall.

The obtained results for our method and the baselines are shown in Table 2. The three benchmark are each representative of a case: In the first, best case, our method outperforms the baselines by a clear margin. In the second case, the performance is slightly superior and/or aligned with the second

Table 2: Performance on global link prediction (higher is better)

| Model | FB15k-237 | | OGBLBioKG | | WN18-RR | |
|---|---|---|---|---|---|---|
| | MAP | F1* | MAP | F1* | MAP | F1* |
| ART (ours) | 0.142 | 0.270 | 0.587 | 0.592 | 0.230 | 0.319 |
| Complex$^2$ | 0.093 | 0.191 | 0.523 | 0.554 | 0.017 | 0.075 |
| Complex | 0.020 | 0.049 | 0.260 | 0.374 | 0.383 | 0.504 |
| NBF | 0.106 | 0.181 | 0.651 | 0.613 | 0.479 | 0.540 |

best baseline, while in the third case the non-probabilistic baselines return better scores. We discuss each case below.

**Case 1: ART clearly outperforms the baselines**. This occurs with `FB15k-237`, which can be arguably considered as the most popular benchmark in the literature. The cost of having a lower overall ranking score of `Complex`$^2$ compared to `Complex` on local rankings, please see Table 5, pays off when looking at the correctness of the global ranking. We confirm, in a different manner, that normalizing the scoring function leads to globally more consistent scores Loconte et al. (2023). However, by not training on artificially created negative samples we can learn a more complex joint distribution.

**Case 2: ART has competitive performance with non-probabilistic methods**. We observe this behaviour with `OGBLBioKg` where our performance is mostly aligned with `NBF`. However, if we look beyond the obtained MAP and F1, which are computed considering true triples in the test set, and focus on the scores obtained for the unknown triples, then we can make an interesting consideration that makes our method a potentially preferable choice. In general, KGE models such as `NBF` are designed for KB completion, i.e., recover the unknown edges, which can be either true or false. Consider the extreme case when all unknown edges turn out to be false. We call it the *pessimistic case*. The opposite is when all unknown triples are indeed true, the *optimistic case*. Neither case is likely to be true. A more realistic scenario is when only a subset is true. To evaluate what happens in each case, we repeated the evaluation of global link prediction on `OGBLBioKG` considering both `NBF` and `ART`. In the optimistic (pessimistic) scenario, all unknown facts above the threshold are assumed to be true (false). In the realistic one, we average the two extreme cases.

The results are presented in Table 3. In the realistic scenario, our method achieves a higher F1 score (with an equivalent MAP), indicating that it is more effective at ranking previously unknown true facts in a more plausible context. One may ask why we did not manually verify the truth of the unknown triples. The challenge lies in the nature of the dataset, which involves biomedical research data, where determining the truth value of triples is often a subject of ongoing research itself.

**Case 3: Local non-generative methods are more suitable.** This case can be seen on `WN18RR`, where the generative models (`ART` and `Complex`$^2$) are clearly outperformed by the discriminative models (`NBF` and `Complex`). There are several reasons for that. Firstly, `WN18RR` is a small KG. The train subset only has $80K$ triples and $11$ relations, which is problematic because generative models need sufficient training data to learn a complex function as the joint distribution while discriminative models need only to determine a truth value based on the local embeddings. Also, it is known that `WN18RR` is a rather regular dataset where there is not much interplay between the entities and relation types. Consequently, methods that limit to learn local scores become competitive also when there is one global ranking. Finally, it has been observed in `WN18RR` that there is a distribution shift between the train and validations sets Loconte et al. (2023), which means that the prior distribution may be misleading. Indeed, we observed only a $1\%$ drop in performance when we artificially cancel out our first two components i.e., $p(s) = 1 \cdot p(r|s) = 1 \cdot p(o|r, s)$.

## 5.3 ON NEGATIVE SAMPLING

Classifiers are commonly learned using negative artificial examples to discriminate true and false edges. This is the case of `ComplEx` and `NBF`. This can lead to a situation when the scores obtained for the true and false triples have different distributions, as it is shown in Figure 2(a). While using

Table 3: `ART` generalises better to unknown triples than `NBF` on `OGBLBioKG`

| Model | Pessimistic | | Optimistic | | Realistic | |
|---|---|---|---|---|---|---|
| | MAP | F1* | MAP | F1* | MAP | F1* |
| ART | 0.587 | 0.859 | 0.840 | 0.592 | 0.723 | 0.719 |
| NBF | 0.651 | 0.796 | 0.788 | 0.613 | 0.723 | 0.700 |

negative (artificially created) negative triples can be beneficial for the predictions, it can also introduce a potentially undesirable bias. A generative model like `ComplEx`[2], as we can see in Figure 2(b) does not lead to such two distinct distributions. However, since it still uses both positive and negative triples for training and the resulting scores of false triples tend to be close to zero, which determines the spike up around those values that we see in the figure. Our model does not learn a distribution of the false edges since this information is not provided to them during training. As a result, we see in Figure 2(c) that although the scores for the negative edges still receive a low probability, they are not excessively concentrated around zero as with `ComplEx`[2], which means our model is more open to the possibility that some of the unknown triples are true.

(a) `ComplEx`  (b) `ComplEx`[2]

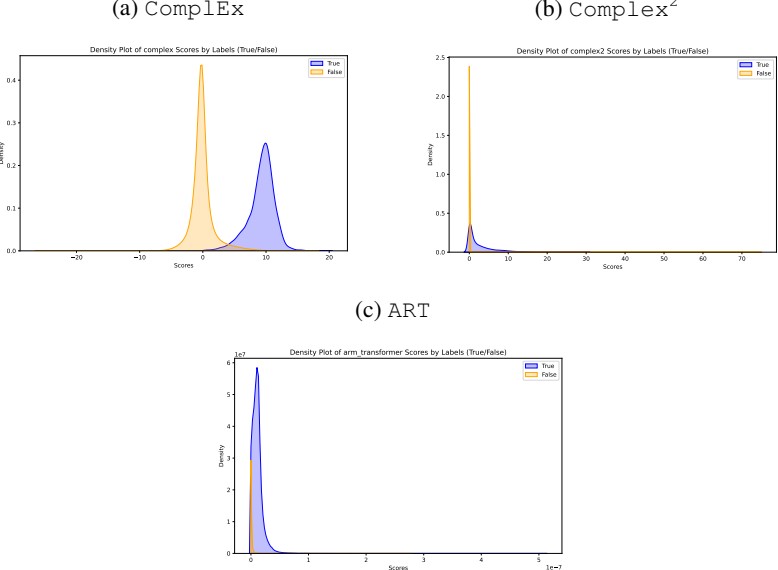

(c) `ART`

Figure 2: Kernel Density Estimation plots of the distributions of true and false triples.

## 5.4 PRIOR ABLATION STUDY

Table 4 reports the MAP scores that we obtain by changing the implementation of the prior distribution ($p(S)$). While a frequency-based distribution is a straightforward choice, our results indicate that it is not always the most effective. Interestingly, the even simpler approach of using a uniform distribution led to better results in two of the three datasets. When we learned the prior jointly with the other components, we were able to obtain even better results, except for `FB15k-237` where the frequency-based prior remained the best choice. These results make it interesting to further study the impact of other prior implementations. This should be seen as a good direction for future work.

## 5.5 LOCAL RANKINGS

For completeness, we also report on the capabilities of our method on *local* link prediction. More specifically, Table 5 reports the obtained MRR on traditional query-based link prediction (the results with $hits@k$ are in Appendix A and B). We emphasise that we do *not* aim to achieve state-of-

Table 4: MAP scores using different priors (higher is better).

| Prior | FB15k-237 | WN18-RR | OGBLBioKG |
|-------|-----------|---------|-----------|
| Frequency | 0.142 | 0.110 | 0.408 |
| Uniform | 0.046 | 0.225 | 0.467 |
| Learned | 0.068 | 0.230 | 0.587 |

Table 5: MRR on local query-based link prediction (higher is better).

| Type | Model | FB15k-237 | WN18-RR | OGBLBioKG |
|------|-------|-----------|---------|-----------|
| non-probabilistic | `Complex` | 0.342 | 0.471 | 0.826 |
| | `NBF` | 0.415 | 0.515 | 0.811 |
| probabilistic (MLE) | `Complex`$^2$ | 0.300 | 0.391 | 0.839 |
| | `ART (ours)` | 0.342 | 0.447 | 0.824 |

the-art performance on this task. Indeed, our results indicate that the non-probabilistic baselines (e.g., `NBF` Zhu et al. (2022)) have superior performance, although our method remains competitive. If we restrict to the probabilistic methods, then our method performs better than the other baseline in two of the three datasets, which makes ours a preferred choice whenever a probabilistic interpretation is required by the use case at hand.

# 6    DISCUSSION AND FUTURE WORK

**Prior** We have shown that it is important to have a prior that resembles the true marginal distirbution p(S), especially when the frequency of the train facts does not resemble the true underlying distribution. We have overcome this by internalizing the prior in the model. However, future work can improve performance by learning more complex priors.

**Compatibility with PDBs** A natural next step in formalizing Complex Query Answering is to use the extensive theory on PDBs Suciu et al. (2011). PDBs require independent Bernoulli variables for every triple, that is, every triple is true or false with probability $p$. When modelling the joint distribution over three categorical random variables, the total probability of all outcomes is one and thus does not meet the requirement of PDBs. However, when the probabilities are learned independently, the scores are not calibrated. Applying min-max normalization to unnormalized scores from a joint distribution, such that every value is between zero and one, and using them as if they represent the score for a triple that is learned independently, led to better-calibrated scores Loconte et al. (2023). Learning independent yet calibrated probabilities to meet the requirements of PDBs remains an open challenge Friedman & den Broeck (2020).

**Increasing the model complexity** We have shown good performance with a stripped down transformer and simple 1D convolutional model on embeddings of rank 150 to emphasise the importance of learning the joint distribution. Future work can increase the rank to 2000 as Complex in this study or increase the model complexity towards the level of NBF to future improve performance.

**One Joint Distribution** A distribution can not be learned if there is insufficient training data. However, we observe that some relations with sufficient training evidence are not picked up at all by both probabilistic methods, `Complex`$^2$ and `ARM`, please see the Appendix C.4. Why some relations are considered out-of-distribution, while there is significant training data, is a question that we leave for future work. Perhaps not all links can be easily captured in a single joint distribution. Next, another limitation of probabilistic methods is that if a link is missing exactly *because* it is out-of-distribution, then it can not be recovered.

To conclude, in this paper we have shown that by sampling the links from a joint distribution $p(S, R, D)$ we can learn consistent probabilities across the KG, with the additional benefit that we no longer need negative sampling for learning. Our results show that, unlike other models,

ours only gives high scores to links for which there is a strong belief they are true. Therefore, the scores can be reliably used for establishing the final truth value of the links, and consequently for Knowledge Base Completion. Moreover, our method returns probabilities that can be used as-is to perform Complex Query Answering systems and thus plugged in existing systems like Complex Query Decomposition (Arakelyan et al., 2021), which require *global* consistency to simple queries.

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

## A  HITS@K

Table 6: Performance comparison of different models on Hits@k. In local rankings, NBF and Complex remain competitive models.

| Model | FB15k-237 | | | | WN18-RR | | | | OGBLBioKG | | | |
|---|---|---|---|---|---|---|---|---|---|---|---|---|
| | H@1 | H@3 | H@5 | H@10 | H@1 | H@3 | H@5 | H@10 | H@1 | H@3 | H@5 | H@10 |
| ART | 0.2491 | 0.3778 | 0.4421 | 0.5302 | 0.3907 | 0.4466 | 0.4724 | 0.5124 | 0.7558 | 0.8722 | 0.9092 | 0.9447 |
| ARC | 0.2422 | 0.3592 | 0.4191 | 0.5003 | 0.3658 | 0.4405 | 0.4678 | 0.5003 | 0.7256 | 0.8493 | 0.8905 | 0.931 |
| NBF | **0.3232** | **0.4555** | **0.5144** | **0.5948** | **0.4971** | **0.5721** | **0.6130** | **0.6616** | 0.7441 | 0.8530 | 0.8926 | 0.9382 |
| Complex | 0.2466 | 0.3693 | 0.4325 | 0.5203 | 0.4330 | 0.4852 | 0.5108 | 0.5445 | **0.7603** | **0.8788** | **0.9163** | **0.9498** |
| Complex2 | 0.2169 | 0.3309 | 0.3889 | 0.4691 | 0.3421 | 0.4233 | 0.4478 | 0.4710 | 0.7744 | 0.8877 | 0.9228 | 0.9538 |

## B  ARC

With a very simple 1D convolutional model with shared parameterization, we are able to get competitive results. This shows the importance of learning the joint distribution, and that it can be learned with a very small model.

Table 7: Local Ranking Performance

| Model | FB15k-237 | WN18-RR | OGBLBioKG |
|---|---|---|---|
| ARC | 0.328 | 0.441 | 0.799 |

Table 8: Global Ranking Performance

| Model | FB15k-237 | | WN18-RR | | OGBLBioKG | |
|---|---|---|---|---|---|---|
| | MAP | F1* | MAP | F1* | MAP | F1* |
| ARC | 0.099 | 0.197 | 0.214 | 0.301 | 0.489 | 0.535 |

## C  REPRODUCE EXPERIMENTS

We use the evaluation mechanism of TorchKGE Boschin (2020).

### C.1  HYPERPARAMETERS

We fix the embedding dimension to $150$, the batch size to $1024$ and smooth the labels with $0.1$. We use the AdamW optimizer. We randomly search the following range of hyper-parameters: lr $\{1e\text{-}3, 1e\text{-}4, 1e\text{-}5, 1e\text{-}6, e\text{-}7\}$, dropout $\{0.4, 0.5, 0.6\}$, weight_decay $\{0.5-2\}$, prediction_smoothing $\{1e\text{-}4, 1e\text{-}5, 1e\text{-}9, 1e\text{-}30\}$. We search the optimal learning rate decrease factor $\{0.1-0.9\}$ after $10$ epochs without improvement.

For ART we search the number of blocks $\{2, 3, 4, 5\}$, number of neurons $\{4, 8\}$. For ART we fix the kernel size to three, and the hidden dimension to $256$.

### C.2  TRAINING TIME

All models are trained within two hours. On OGBLBioKG it is competitive within two hours, but converges within 8 hours. Faster than NBF, but slower than Complex[2]

## C.3 STATISTICAL SIGNIFICANCE LINK PREDICTION

We test the statistical significance of our best model, ART, and assume the same underlying probabilistic process for ARC. We run with 10 random seeds on FB15k-237 and WN18-RR. Resulting in two standard deviations, 0.00124, 0.00917, respectively. Due to the small differences, we decided to omit this test on OGBLBioKG for environmental reasons because the models take longer to train on this dataset.

## C.4 MRR PER RELATION

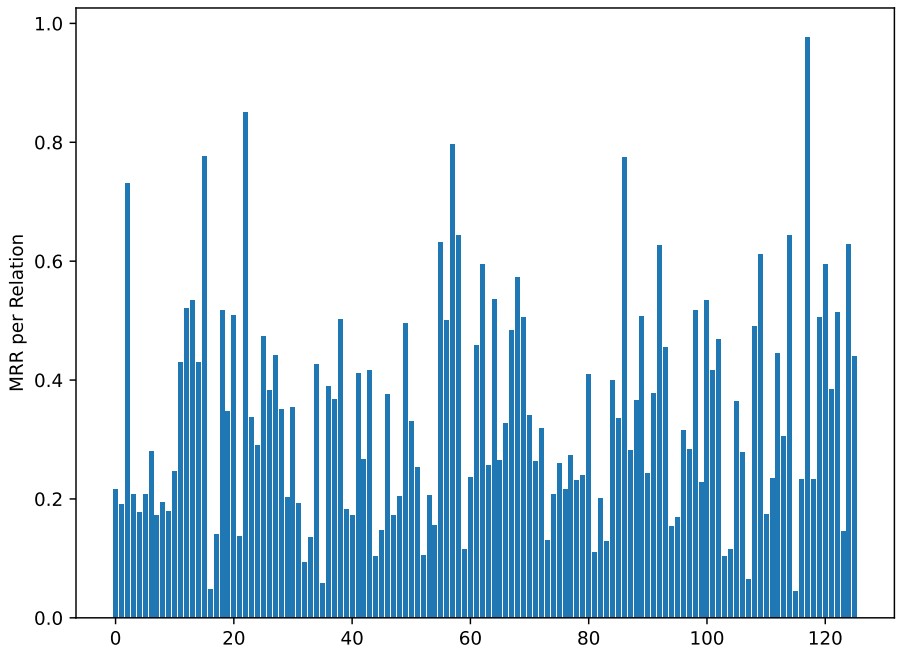

Figure 3: Plotting the MRR per relation on FB15k-237 to observe the difference in performance per relation type. We filter at least 20 facts per relation because MRR for a low number of facts can lead to extreme scores. The relation IDs are not specified because this figure only aims to visualise the high variance of MRR per relation in general
.

