# OpenReview forum: "AutoRegressive Knowledge Base Completion"
_ICLR.cc/2025/Conference — Submitted to ICLR 2025_

### Official Review · Reviewer_NDxE · 2024-10-25

**Soundness:** 2
**Presentation:** 2
**Contribution:** 2
**Rating:** 3
**Confidence:** 5

**Summary:**

1. This paper presents a generative model for knowledge base completion that learns the distribution of triples $p(S, P, O)$ through an autoregressive approach, where $p(S, P, O) = p(S) p(P|S) p(O|S, P)$, optimized via maximum likelihood estimation (MLE).

2. Experimental results show that the model achieves competitive performance across three knowledge graphs (WN18RR, FB15k-237, and OGBLBioKG), demonstrating its effectiveness for tasks requiring consistent truth values, such as complex query answering.

**Strengths:**

1. This work offers two notable advancements over traditional KGE methods: a) it eliminates the need for negative sampling, which improves training efficiency; and b) it provides a probability score for each query, allowing for comparisons between distinct queries.

2. Empirical results regarding negative sampling reveal insightful perspectives, showing that this approach can introduce a potentially undesirable bias in training.

**Weaknesses:**

1. Presentation Issues

    1.1. Inappropriate language usage: Frequent grammatical and spelling errors detract from readability.

    1.2. Insufficient figure and table descriptions: Missing descriptions make it unclear what each component represents.

    1.3. Inadequate method explanation: The ART model lacks a diagram, and the structural and mathematical details of both the ARC and ART models are insufficient.

2. Experimental Concerns

    2.1. Reproducibility issues: Due to incomplete model descriptions, achieving reproducibility is challenging.

    2.2. Missing ARC results in Table 2: Including ARC results alongside other methods would allow for a more comprehensive comparison.

    2.3. Weakly supported conclusions: The claim on Page 8, Line 395, that “our model is more open to the possibility that some of the unknown triples are true,” is not backed by experimental evidence. How does this openness benefit the model?

**Questions:**

Setting aside the weaknesses mentioned, my primary concern lies with the overall contribution of this work. The primary objective of KGE methods is to complete missing triples in KGs. Models like $\text{Complex}^2$ proposed by [1] aim to adapt existing KGE methods into generative models, enabling applications in maximum likelihood estimation (MLE) and sampling. However, the authors emphasize that generating new triples—a core capability of generative models—is not necessary for this work. Additionally, the knowledge graph completion performance of this model, as shown in Table 6, is relatively underwhelming. In summary, the model seems unable to learn a robust distribution over triples or achieve strong knowledge graph completion. So, what is the intended purpose of this model?

[1] Loconte, L.; Di Mauro, N.; Peharz, R.; Vergari, A. How to Turn Your Knowledge Graph Embeddings into Generative Models. Advances in Neural Information Processing Systems 2023, 36, 77713–77744.

---

> ### Author Response · Authors · 2024-11-22
>
> Thank you for your review
>
> 2.1
> Every experiment is fully reproducible (see the README in the supplementary material). We can upload a link to the trained models if you wish to redo the experiments without retraining. In Appendix Section C1, we report the range of hyperparameters we used. However, we agree that the paper should be self-contained, without relying on the supplementary material. In the next version, we will address this.
>
> 2.2
> We moved the results of ARC to the Appendix at the last minute because both ART (Transformer) and ARC (1D Convolutions) model the same distribution and are both Deep Autoregressive Models. However, ART performs better, which is why we kept it in the main paper and placed ARC in the Appendix. We included ARC to emphasize that we can already learn an expressive distribution with simple 1D convolutions, rather than relying solely on deeper/modern networks.
>
> 2.3
> We actually show this with experimental evidence. Please see Table 3. For both ART and NBF, we optimize the F1 score (balancing precision and recall) by considering all possible thresholds on the F1 confidence curve. Triples above the threshold are classified as true, and those below as false. This simulates the procedure of completing the knowledge base for all possible triples.
>
> Evaluating this procedure works well in the setting of labelled data. However, for KGs, we usually don’t know the values of unknown triples. We assume that all unknowns are false and check if the withheld triples are ranked higher than the unknowns. This assumption is useful for evaluation but not entirely realistic: if all unknown triples were false, we wouldn't need KGE models in the first place, as it's exactly the true unknown triples that we aim to recover.
>
> We therefore experiment with the idea of "what if the model is right," by considering all unknown triples with high probability (i.e., above the threshold) as true instead of false. We call this the optimistic scenario. Since we have more triples with high probability, this significantly improves our performance compared to NBF. We agree that this analysis is arbitrary, but note that simply assuming all unknown facts are false (as we currently do in Link Prediction and KBC) is also definitely incorrect. Therefore, we also present a realistic scenario that averages both metrics.
>
> Q:
> Similar to Complex², we model the joint distribution over three categorical variables $p(S,R,O)$.
> Learning this distribution has been recognized by the community as a promising direction.
> However, we argue that their efficient normalisation technique only works for a certain class of KGE models with simple scoring functions (e.g., Complex uses the dot product on the embedding vectors). We address this limitation by starting from deep KGE models e.g., ConvE, which allows for more expressiveness in the scoring function. To emphasise that for generative modelling, simple scoring functions are not sufficient, we keep the overall size of our models small  (ARC: 8M, ART: 10M, Complex²: 59M). Yet, we are able to learn a more expressive joint distribution, which we show by reporting better Link Prediction and Knowlegde Base Completion performance.
>
> Additionally, we show that even if the reasoning task is slightly more advanced than link prediction, we already outperform the current state of the art on link prediction if the KG is complex enough (FB15k-237). However, note that learning one joint distribution is a more challenging task than classifying triples. Thus, we do not aim to beat them on all benchmarks, although we give a possible explanation in the general rebuttal above.
>
> We consider the truthfulness of a triple to be the likelihood that it will be sampled from the joint distribution; hence the scores are globally consistent by design.
>
> Naturally, we can also “sample” new triples from our model, however, in the current setup, all relations and entities are known, so simple inference suffices. Yet, in more advanced tasks, like determining the most likely graph out of all possible graphs $p(G)$, we could use our model to generate the most likely graphs by sampling from our distribution over triples, narrowing down the search space significantly. Yet, we consider this future work.
>
> Presentation issues.
>
> We agree with all presentation concerns. If we have convinced you of our contribution and the significance of our results, we can upload a new version addressing all presentation concerns and incorporating your feedback.

---

### Official Review · Reviewer_NQty · 2024-10-27

**Soundness:** 1
**Presentation:** 2
**Contribution:** 1
**Rating:** 3
**Confidence:** 4

**Summary:**

Paper Summary:
The paper proposes a novel autoregressive approach for Knowledge Graph (KG) completion, aiming to address limitations in traditional Knowledge Graph Embedding (KGE) methods, which often rely on local scoring and negative sampling. Instead, the authors introduce an autoregressive generative model, which learns a global, joint distribution across entities and relations in the KG without requiring negative sampling. By using neural networks to model conditional probabilities, this approach produces interpretable global scores for query answering.

Conclusion:
The paper proposes a novel autoregressive approach for Knowledge Graph (KG) completion, which learns a global, joint distribution across entities and relations in the KG without requiring negative sampling. Although the proposed approach is novel and interesting, its performance is less convincing due to lack of state-of-the-art results, lack of baseline models and lack of benchmark datasets. So in general, this is not an acceptable paper to ICLR.

**Strengths:**

Strength:
1. The method proposed by the authors are relatively novel. The introduction of an autoregressive generative model to address these issues by learning a joint distribution across entities and relations without negative sampling is compelling. This approach yields global, interpretable scores, which appears to enhance the method's adaptability to large KGs.
2. The method and deep network structure is described with details in this paper. Conditional probability reasoning is described with detailed mathematical formats. The paper is well-organized as well.

**Weaknesses:**

Weakness:
1. The performance of the model is not convincing.
(i): The F1 scores for the proposed model is not comparable to NBF on the WN18-RR dataset. Also, the performance of the model on the OGBLBioKG is several percentages behind the NBF model. So in general, the proposed ART model only out-performs the baselines on the FB15K-237 dataset, which is not convincing.

(ii): Other than NBF and Complex, there are so many other baseline models published in the recent years: RotatE, GPFL, ConvE, DisMult, SACN, DRUM, Neural-LP, CoMPILE, etc. In order to make the paper more convincing, the authors should definitely include more baseline models when evaluating the performance of the proposed model.

(iii): I am not familiar with the OGBLBioKG dataset, which I assume is a biomedical knowledge graph dataset. But now that working with bio KG, why not use Bio2RDF and DRKG, which are more popular benchmark datasets? Also, I think most link prediction researchers will use WN18-RR, FB15K-237, Nell-995, YAGO3-10, DBpedia (50K & 500K). Why not include Nell-995, YAGO3-10, DBpedia? I am wondering why the authors only select specific KG reasoning datasets to evaluate their model. This issue makes the paper less convincing.

2. Some minor defects:
While the paper mentions applications in complex query answering and knowledge base completion, specific examples of query types and scenarios could illustrate how the proposed model would outperform traditional KGEs. A few comparative examples could significantly enhance the paper's practical utility.
Figure 1 is too small, could you please enlarge the font size?

**Questions:**

1. Why not include more baseline models?
2. Now that working with bio KG, why not use Bio2RDF and DRKG, which are more popular benchmark datasets?
2. Now that working with WN18-RR and FB15K-237, why not include Nell-995, YAGO3-10, DBpedia?

---

> ### Author Response · Authors · 2024-11-22
>
> Thank you for your review.
>
> W1 - Performance against baselines
> We do not agree that the performance is unconvincing. Similar to Complex², we aim to model the joint distribution over triples with three categorical random variables according to the exact MLE objective. By directly addressing a limitation of this model, we outperform Complex² on Link Prediction and Knowledge Base Completion while using significantly fewer parameters.
>
> Generative modeling is inherently a more challenging task than classification. Please refer to the general rebuttal at the top of this page for a possible explanation as to why we do not outperform NBF on knowledge base completion.
>
> We included NBF to (1) validate wheter we are competitive on link prediction, and (2) to emphasise that we can already outperform the state-of-the-art if the reasoning task is slightly more advanced than Link Prediction, namely knowledge base completion, given that the KG is sufficiently challenging (FB15k237).
>
> We include Complex as the model class for KGE models that model the distribution $p(Y_{sro}∣S,R,O)$. While I am not familiar with all the models you mentioned, most KGE models, such as RotatE, DistMult, and ConvE, can be interpreted as modeling each triple with a separate random variable. Also, note that although Complex scores higher on Link Prediction, its identical but normalized variant, Complex², performs better on Knowledge Base Completion. This validates, in a different way, that the scores of Complex² are more globally consistent and that the global ranking serves as a measure of consistency. This is further supported by the fact that we already know Complex² is better calibrated, as detailed in their section on calibration in the Appendix.
>
> Datasets
> We include FB15k-237 and WN18RR as they are standard benchmarks for Link Prediction, with many top conference papers using only these datasets over the years. We also include OGBL-BioKG because both NBF and Complex² have used this dataset, allowing for a fair comparison. Without this, we would need to optimize their models (e.g., grid search), and KGE models are very sensitive to hyperparameter tuning. While we agree that adding more benchmarks and models would strengthen the paper, we believe the current models, benchmarks, and evaluation already effectively demonstrate our key points:
>
> We learn a more expressive joint distribution than our baseline, Complex², while using fewer parameters. If the reasoning task is slightly more advanced, we have already surpassed the state of the art on link prediction when the knowledge graph is sufficiently complex, all with simple models.
>
> We believe WN18RR is flawed and should no longer be the standard benchmark in the community.
> Although Complex is popular and effective for Link Prediction, it cannot reliably be used for more advanced reasoning tasks.
>
> Presentation
>
> We included an example of different queries in the general rebuttal above, but we agree that an example in the paper would have been more informative. We will add this in the next version. If we have convinced you of the contribution and significance of the results, we can upload a revised version with an improved presentation and your feedback (e.g., with the example) in a few days.

---

### Official Review · Reviewer_BLUw · 2024-11-04

**Soundness:** 2
**Presentation:** 1
**Contribution:** 1
**Rating:** 3
**Confidence:** 4

**Summary:**

This paper proposes ART and ARC, two auto-regressive generative models that learn a joint distribution of the entities and relations of the KG without resorting to negative sampling.

**Strengths:**

1, The proposed method does not need negative sampling in the training.

2, The proposed method give consistent global scores instead of local scores.

**Weaknesses:**

1, The novelty is not clear in this paper. The proposed autoregressive model is a simple use of the conditional probability formula $p(s,r,d)=p(s)*p(r|s)*p(d|r,s)$ which lacks novelty. Can the authors explain more about the novelty of this paper?

2, The empirical performance of the proposed method is not strong. In Table 2, compared with NBFNet, ART only shows slightly better performance on FB15k-237 dataset while being outperformed by a large margin on OGBLBBioKG dataset and WN18-RR. In Table 3, ART did not show convincing advantages neither. The emprical results are not enough to show the superiority of the proposed method. Can the authors design other experiments to show the strength of their methods?

3, This paper needs to discuss more on the difference of 'local' and 'global' scores. Learning 'global' scores is the contribution claimed in the paper. However, the concepts of 'local' and 'global' are not clearly defined. What's the necessity of computing global scores in the knowledge base completion tasks and downstream tasks? What advantages do global scores have over local scores especially give that the empirical results of global score methods are even worse than existing local score methods?

4, The presentation of the paper is not good. (1) Figure 1 shows the architecture of ARC instead of ART. However, ARC is not even in Table 2 or Table 3. (2) Best numbers in Table 2 and Table 3 are not highlighted. (3) The legends of Figure 2 are too small and can not be seen clearly. (4) Figure 4 does not have a name for X-axis.

**Questions:**

Please refer to the questions mentioned in the Weaknesses part.

---

> ### Author Response · Authors · 2024-11-22
>
> Thank you for your review
>
> W1
> As stated in the related works, many models indeed autoregressively model the likelihood of a triple. However, we model the joint distribution, $p(S,R,D)$, using only three categorical random variables, following the exact MLE objective, similar to Complex². To the best of our knowledge, this is the only method that models this joint distribution exactly. However, they normalise the output of tensor factorisation methods by efficiently calculating $Z$. A limitation of this approach is the limited expressiveness in the scoring function of those class of models.
>
> By starting from a deep KGE method e.g. ConvE, and turning it generative, we have more expressiveness to learn a complex function such as the joint distribution. Deep methods are known to be parameter-heavy. To emphasise that we directly address this limitation, we keep the overall size of our models small (ARC: 8M, ART: 10M, Complex²: 59M). This contribution should be a direction for further research. E.g., in Complex², the mixture weights of the underlying Probabilistic Circuit are assumed to be constant, which might limit expressiveness. Also, future work could increase the depth of ART/ARC even further to close the gap on the task of Link Prediction with the state-of-the-art.
>
> W2.
> We do not agree that the empirical evidence is not strong. We show that we learn a more expressive joint distribution than Complex² by reporting better performance on Link Prediction and Knowledge Base Completion across all benchmarks. Note that modelling one joint distribution is more challenging than classifying triples. We included NBF, a discriminative classifier that is state of the art on Link Prediction, in our analysis to (1) validate that we are competitive on Link Prediction and (2) to emphasise that we can already surpass this model when the reasoning task is slightly more complex, namely Knowledge Base Completion given that the benchmark is sufficienlty complex (FB15k237).
> Moreover, eventhough the global ranking performance of NBF is arguably higher, it is still a ranking task, therefore this does not nessecarcy imply that their disriminative scores are more preferred for reasoning. Although we give a possible explanation in the general rebuttal above on why we do not outperform them on this benchmark, this is not our aim. We try to learn a more expressive joint distribution, an inherently more challenging task than classifying triples.
>
> W3
> We introduce these terms to emphasize that traditional KGE models may not have a consistent belief regarding the truthfulness of a triple. As shown in the general rebuttal above, we use "local" because each query for link prediction is evaluated separately and then aggregated. By first collecting all facts and evaluating the "global" ranking, we demonstrate that if the model is inconsistent, the scores for link prediction do not necessarily imply a correct "global" ranking. Note that Complex and Complex² are identical models, but Complex² is normalized and probabilistic. Complex outperforms Complex² on link prediction (local), yet Complex² scores higher on knowledge base completion (global). Learning globally consistent scores is indeed a more challenging task, which is reflected in the lower link prediction performance of Complex² compared to Complex.
>
> W4.
> We fully agree. Some last-minute changes negatively impacted the overall presentation, though this is fixable. If we have convinced you of our contribution, we can upload an improved version in a few days, addressing all presentation concerns.

---

> > ### Comment · Reviewer_BLUw · 2024-11-26
> >
> > Dear authors,
> >
> > Thanks for your answers. However, I will keep the current score because I think the paper needs further polishing to improve the soundness of the proposed methods .
> >
> > Best,
> > Reviewer BLUw

---

### Official Review · Reviewer_327F · 2024-11-06

**Soundness:** 1
**Presentation:** 1
**Contribution:** 2
**Rating:** 3
**Confidence:** 4

**Summary:**

The paper introduces an autoregressive model for knowledge graph (KG) completion that avoids negative sampling. Traditional knowledge graph embedding (KGE) models rely on negative sampling, which can be problematic under the Open World Assumption (OWA), as distinguishing true facts from false negatives can lead to inconsistent scoring. The proposed approach aims to avoid this by factorizing the joint distribution $p(S, R, D)$  into $p(S) p(R \mid S) p(D \mid S, R)$  and modeling each factor as a categorical distribution.

While the problem addressed is significant and the approach of using an autoregressive model to learn effectively under the OWA without relying on negative sampling is novel and interesting, there are concerns regarding the proposed solution's modeling choices, the experiment setting, the fairness and soundness of the evaluation strategy, and the lack of convincing empirical results.

**Strengths:**

1. **Addressing the Open World Assumption**: The paper addresses an important gap in existing KGE methods: the Open World Assumption (OWA). This challenge is particularly relevant for KGs, as real-world KGs are highly incomplete, and it cannot be assumed that unseen facts are false. Therefore, an ideal training methodology should respect the OWA, unlike common negative sampling methods, which incorrectly treat unseen facts as negatives.

2. **Maintaining Probabilistic Interpretation during Training**: The proposed probabilistic factorization modeling and autoregressive approach is notable for maintaining a global probabilistic score throughout training—something not guaranteed by most existing methods. Typically, existing approaches either maintain a local probabilistic score per query or use a global energy-based score that requires time-consuming post-hoc normalization.

**Weaknesses:**

**W1. Unclear Presentation of Existing Literature Gap**: The paper could benefit from a clearer presentation of the gaps in existing literature, an explanation of why addressing these gaps is important, and a more intuitive explanation of how the proposed method addresses these challenges. To my understanding, there are two main gaps in existing KGE methods that this paper attempts to address: (1) the Open World Assumption (OWA) challenge and the use of negative sampling, which conflicts with the OWA, and (2) the lack of a global probabilistic score consistent throughout training. The second gap can be further divided into two cases: (2.a) some query-based methods (e.g., NBFNet) can maintain a probabilistic score, but these scores are conditioned on the query and not global, and (2.b) other global KGE methods are mostly energy-based, requiring post-hoc global normalization, which can be time-consuming. Addressing the first gap is crucial, but the paper does not clearly articulate why solving the second gap is important. The introduction could be improved with a deeper explanation of why addressing these two gaps matters, perhaps by including a figure for illustration or providing a concrete example to support the argument.

**W2. Missing Discussion of Related Work and Calibration Techniques**: One of the key promises of the proposed method is its ability to maintain a consistent global probabilistic score that is informative for downstream decision-making by accurately reflecting the likelihood of an unknown fact being true. This is a well-studied concept in the literature known as calibration. Calibration is crucial because it enables determining an optimal threshold for making predictions. Without proper calibration, the probabilistic scores learned by a neural network may not align with real-world probabilities, as the scores are optimized solely to satisfy the training objective. This could lead to arbitrarily high scores for positive training examples and arbitrarily low scores for negative examples, compromising reliability.

Although the paper cites Zhu et al. (2023) (on Line 62), it lacks a broader discussion on calibration techniques and their relevance to the proposed approach (e.g., [1] for KGE and [2] for general classifiers). More significantly, the proposed methodology does not implement any common calibration techniques, which undermines the claim that the learned global probabilistic score is consistent and useful for downstream decision-making. Including a discussion of calibration techniques and incorporating such methods into the proposed approach would strengthen the validity of this claim.

**W3. Problematic Modeling Choice**: The proposed method circumvents negative sampling by decomposing the joint distribution $p(S, R, D)$ into $p(S) p(R \mid s) p(D \mid S, R)$ and assumes $p(R \mid s)$ and $p(D \mid S, R)$ are categorical distribution by applying Softmax activations. However, this modeling assumption is problematic because it suffers from poor expressivity. There is a reason why most existing KGE choose to model each individual edge as a Bernoulli random variable, rather than modeling $p(D \mid S, R)$ as a categorical distribution. Specifically, a categorical $p(D \mid S, R)$ assumes that given a particular $S=s$ and $R=r$, there is one and only one target entity $D=d$. This is generally not true in real-world KGs. For instance, in a KG about company structure, a single source entity may link to multiple target entities with the same relation, as seen in triples like (Google, employee, Jeff Dean), (Google, employee, Salar Kamangar), (Google, employee, Craig Silverstein). Even if the KG is carefully constructed to avoid such one-to-many mappings, the proposed method introduces inverse triples (LIne 251), converting any many-to-one mapping to one-to-many in the inverse, which violates the categorical assumption. The only scenario where the proposed method does not suffer from poor expressivity is when all relations are unique one-to-one mappings—an unrealistic scenario for most real-world KGs.

Another problematic assumption mentioned in the paper is that the dataset is sampled i.i.d. (Line 134). This assumption does not hold for KGs because entities are shared among different facts. For example, if (Google, employee, Jeff Dean) is observed in the dataset, then it is far more likely that (Google, owned by, Alphabet Inc.) will also be observed, then in a dataset where (Google, employee, Jeff Dean) is not observed, because perhaps the previous case is a KG about company structure, whereas the latter is a KG that has nothing to do about Google. The paper could greatly benefit from either improving the methodology to circumvent these assumptions or, at the very least, including a discussion on why these assumptions can be made and why their violation might not significantly affect practical outcomes.

**W4. Strange Experiment Desing and Unfair Evaluation Against Baseline Methods**: The experiment task was "global link prediction" (Line 281). However, this task is not designed in such a way that could showcase the potential benefit of the proposed method. As the proposed method is supposed to fix the gap of OWA challenge in existing KGE methods and thus could produce a "consistent global probabilistic score", a better experiment setting could be to first define a measure of what it means for a model's score to be "consistent" for the given task at hand, and then evaluating and showcasing that the proposed autoregressive model indeed outputs scores that are more "consistent" than the baseline methods, while achieve on-par performance to the baseline on the link prediction task. In contrast to simply attempting to beat SOTA methods on link prediction performance, such experiment can truly highlight the competitive edge of the proposed method. Regarding how to define such a "consistency" measure, a suggestion is to take inspirations from the calibration literature ([1][2]) and use their experiment methodology to show how the calibrated score is more informative to downstream users.

On the other hand, the link prediction performed "globally" is a task whose evaluation is unfair to query-based methods such as NBFNet, because they are designed solely for query-based prediction. If one is only interested in asking whether an unknown triple $(s, r, d)$ is true, then one should apply the query-based method as they were designed, which is to look at the score of all potential $d$'s by giving a specific $s$ and $r$. By "executing all queries from the test set and merging the obtained scores together" (Line 313), one greatly diminishes the expressivity of NBFNet, which was exactly one of the original challenges that NBFNet set out to address.

The authors argue that the "global link prediction" task is valid because one prefers global scores to local scores, because local scores hinder one's ability to "support more effectively complex and inter-contextual query answer and to perform KB completion across the entire scope of the KB" (Line 282 -284). This is not true, as many work has studied using query-based method like NBFNet to perform complex query answering on KGs. For instance, Zhu et al. [3] first proposed a method named GNN-QE, which is essentially applying fuzzy logic operators onto NBFNet to effectively and efficiently perform complex logical query answering (CLQA) on KGs. A follow-up work [4] then showed that GNN-QE can perform CLQA while generalizing to new KGs with unseen entities. Finally, a most recent work from Galking et al. [5] proposed the method ULTRA-Query that is able to generalize to KGs to different domains with completely unseen entities and relations types. Hence, these literature has solved the challenge of using query-based KGE models for effective and inter-contextual query answering task.

**Questions:**

Q1. **Alternative modeling choice**: Would it be possible to consider other probabilistic models instead of categorical distributiosn on $p(R \mid S)$ and $p(D \mid S, R)$? Say, for example, a mixture of distributions where it is allowed to have at most $K$ modes (e.g. peaks in the p.d.f. curve) for some predefined hyperparameter $K$? This way, we might be able to model multiple ground-truth $D$’s given specific $S$ and $R$ and (to some extent) cope with the one-to-many relational mappings in real-world KGs. Another way to think about this potential improvements is to draw analogy to multi-head attentions. If we interpret the Softmax logits as attention scores over the vocabulary of entities, then having multi-headed attention is essentially implementing a mixture of categorical distributions. Would it be possible to improve the proposed methods with multi-headed “attentions”?

Q2. **Conflicting Experiment Results**: In Section 5.3 and Figure 2, it is explained that “we see in Figure 2(c) that although the scores for the negative edges still receive a low probability, they are not excessively concentrated around zero …” However, looking at Figure 2(c), it seems that the negative edge score (orange curve) is indeed excessively centered around 0. Could you explain this discrepancy between the Figure and the text?

**References**

[1] Tabacof, Pedro, and Luca Costabello. "Probability calibration for knowledge graph embedding models." (2019).

[2] Silva Filho, Telmo, et al. "Classifier calibration: a survey on how to assess and improve predicted class probabilities." (2023).

[3] Zhu, Zhaocheng, et al. "Neural-symbolic models for logical queries on knowledge graphs." (2022).

[4] Galkin, Michael, et al. "Inductive logical query answering in knowledge graphs." (2022).

[5] Galkin, Michael, et al. "Zero-shot Logical Query Reasoning on any Knowledge Graph." (2024).

---

> ### Author Response · Authors · 2024-11-22
>
> Thank you for your review.
>
> W1
>
> A discriminative model requires negative examples to differentiate true from false, making negative sampling strategies important. In contrast, generative models can't model the distribution of unknown data. For example, we can learn the distributions of cats and dogs from images, but without knowing what unknown edges are false, we can't learn their distribution. Therefore, we focus only on known triples (train-KG).
>
> KGE methods treating each edge as an independent random variable $p(Y_{sro}​∣S,R,D)$ are unnormalized, resulting in inconsistent scores unsuitable beyond LP. Normalizing this distribution to learn $p(Y,S,R,D)$ would require calculating the normalization constant Z (Complex² ) or decomposing this joint distribution into valid conditional probabilities (ART/ARC). However, for the reason mentioned above, we discard the labels.
>
> Post-training normalization yields valid probabilities but limits the model’s ability to learn the true data distribution. All generative models learn the joint distribution during training.
>
> W2
>
> We consider post-calibration methods an interesting but orthogonal research direction. While we do not implement a specific calibration technique, our approach inherently produces calibrated scores by learning the joint distribution over three random variables. If the total probability mass equals one, applying min-max normalization or a sigmoid ensures values fall within [0,1] (See Callibration section in Complex²).
>
> We do not use Expected Calibration Error (ECE) due to fundamental challenges with the Open World Assumption in KBC. ECE introduces systematic bias because high-probability bins are unfairly penalized when correctly assigning high probabilities to facts that are true but unlabeled. This bias affects calibration evaluations more severely than KBC.
>
> W3
>
> It seems there may be a misunderstanding regarding the principles of generative modelling. While it is correct that replacing the Sigmoid with Softmax in a discriminative model like ConvE would violate the categorical assumption, we would like to clarify that our approach is not focused on edge classification $p(Y_{ro}∣R,D)$; instead, we aim to learn the underlying data distribution $p(S,R,D)$.
> The i.i.d. assumption in ARMs applies to the samples used for Monte Carlo sampling, where each sample is treated as independent.
> However, this is distinct from the assumption about the structure of the underlying dataset (i.e., the KG). The dataset itself may have correlations (as is typical in KGs), and ARMs are designed to model such correlations. In fact, we explicitly try to model these dependencies.
> I hope this helps clarify the distinction.
>
> W4
>
> We agree that "merging the obtained scores" was a poor choice of wording. What we actually did was what you suggest, we evaluate all potential $d$'s for a given $s$ and $r$, as clarified in the general rebuttal above. We then collected and ranked all these facts into a single “global” ranking, simulating the process of actual KBC. This task is slightly more advanced than simple LP, as it requires scores to be consistent across all queries.
>
> Note that Complex achieves higher MRR on LP but lower MAP on KBC compared to its probabilistic counterpart, ComplEx². This supports our choice of MAP as a metric for consistency, as Complex² is better calibrated (see the section on calibration in Complex²). Moreover, on FB15k-237, we show that even with simple architectures, our method produces a more consistent global ranking than the SOTA on LP.
> Is this metric perfect for evaluating consistency? No. As seen in OGBLBiokg, discriminative methods can achieve high scores if true and false values are widely separated, pushing all true values to the top of the global ranking.
> Therefore, I am happy that you state that GNN-QE is suited for CQA, because very recent work (https://arxiv.org/abs/2410.12537) shows that most queries in CQA can be reduced to 1p queries. And there is no need for consistent scores, as the truth value of the query only depends on the “local” LP scores.
>
> When they adjust for this, the GNN-QE scores are very poor. Hence, we argue that CQA is not “solved”. However, we do see the use of generative models for CQA as separate work. In this work, we focus on learning consistent scores.
>
> Q1: We appreciate your suggestion to increase expressivity by adding mixtures or additional attention heads. In this work, however, we intentionally kept the overall number of parameters low to highlight and address the limitations of Complex², specifically its limited expressiveness in the scoring function.
>
> Q2: We agree that this figure is not sufficiently informative and will omit it in the next version.
>
> We fully agree with your concerns regarding the presentation. If we have addressed your concerns about our contribution, we can upload a revised version with improved presentation incorporating your feedback in a few days.

---

### Author Response · Authors · 2024-11-22

We appreciate the reviewers' feedback and recognize that overlapping concerns about the relevance of our work suggest we may not have communicated our contributions as effectively as intended. To clarify, our paper addresses the limitations of Complex², the only KGE model, to our knowledge, that uses exact MLE to model the joint distribution p(S,R,D). We demonstrate that our approach captures a more expressive joint distribution, resulting in superior performance on Link Prediction (LP) and Knowledge Base Completion (KBC).

While neither our method nor Complex² achieves state-of-the-art LP performance, modeling $p(S,R,D)$ is already appreciated by the community as a promising direction. This is evident in the recognition of Complex², which was accepted as an oral presentation at NeurIPS (top 0.6%)—and rightfully so, as it’s a very well-written paper.

The authors propose an efficient method for calculating the normalization constant, but it is restricted to tensor factorization models with simple scoring functions, e.g., the scoring function in Complex is a dot product. While sufficient for discriminative models $p(Y_{sro}​∣S,R,D)$, such a scoring function lacks the expressiveness needed for generative tasks $p(S,R,D)$.

We propose an alternative approach that starts with deep KGE models, which offer more expressive scoring functions. By adapting a discriminative deep KGE model like ConvE to a generative one, we introduce ARC. This is achieved by replacing the Sigmoid with a Softmax, removing artificial negative examples, and adding two components, $p(s)$ and $p(r∣s)$, ensuring valid probability distributions without explicit normalization. This simple yet effective solution provides consistent probabilities while capturing more complex dependencies through a richer scoring function. Additionally, we also introduce ART, which follows the same approach but uses a transformer instead of convolutions. Deep KGE models are known to be parameter-heavy, but we intentionally keep our models small (ARC: 8M, ART: 10M, Complex²: 59M) to emphasize the relative increase in parameters within the scoring function, rather than overall model size. This highlights our contribution and encourages further research. For example, the assumption of constant mixture weights in the Probabilistic Circuit underlying Complex² could limit its expressiveness.

LP is evaluated per query, meaning perfect MRR can be achieved without globally consistent scores. For instance, consider a test KG with triples [(d1, r1, s1), (d1, r2, s1), (d2, r2, s2)]. For q1​, rank(d | s1, r1), the model ranks [d1=99, d2=88, d3=87], achieving perfect MRR since the true triple ranks first. Similarly, for q2​, rank(d | s1, r2), it ranks [d1=33, d2=32, d3=31]. While both "local" rankings are correct, the global ordering [99(T), 88(F), 87(F), 33(T), 32(F), 31(F)] shows false triples from one query scoring higher than true triples from another. This demonstrates that for simple queries, only the ranking is reliable/measured, while the scores can lack consistency, limiting their reliability for any other reasoning task.

On FB15k-237, we demonstrate this phenomenon by surpassing the state-of-the-art LP method, NBF, by 4% MAP (global), despite NBF achieving 7% higher MRR (local). This highlights the importance of learning globally consistent scores. We included NBF in our analysis to show that, when the KG is sufficiently complex, we can already surpass the SOTA in LP if the reasoning task is slightly more advanced. However, our goal is not to outperform NBF on all benchmarks. Our goal is to learn a more expressive joint distribution, which is inherently a more challenging than classifying triples. Still, we offer a possible explanation for why we do not surpass NBF on other benchmarks.

For WN18RR, the distribution shift between train and test KG, along with limited entity interactions, highlights the benchmark's limitations. We suggest it should no longer be the KBC standard. On OGBLbioKG, all models achieve similar high performance (~0.83 MRR), easily distinguishing true from false triples. For example, rankings like rank(d | s1, r1) = [d1=100, d2=1, d3=0] and rank(d |s1, r2) = [d1=100, d2=100, d3=0] lead to a global ranking of [100(T), 100(T),100(T), 1(F), 0(F)], showing that when the difference between true and false edges is large, even discriminative models can achieve perfect global rankings. While global ranking is more challenging due to the need for consistency, perfect performance can still be achieved, even with a discriminative model.

We did not evaluate on Complex Query Answering (CQA) due to additional domain-specific challenges. While recent work (https://arxiv.org/abs/2410.12537) showed that many CQA queries can be reduced to LP, when the benchmarks were limited to non-reducible queries, their performance was actually poor. This reinforces the need to first focus on foundational improvements, like globally consistent probability scores.

---

### Meta-Review · Area_Chair_PqqR · 2024-12-18

**Metareview:**

This paper proposes an autoregressive generative model (ART/ARC) for knowledge graph completion that learns a joint distribution of entities and relations without using negative sampling. The key contribution is modeling P(S,R,D) directly using categorical distributions and maximum likelihood estimation, with the goal of producing globally consistent probability scores that can be compared across different queries. While the approach addresses important limitations regarding the Open World Assumption and negative sampling, and maintains globally consistent probabilistic scores throughout training, the technical implementation falls short in several aspects.

The main weaknesses include limited empirical validation, with the proposed models only outperforming baselines on one dataset (FB15k-237) while underperforming on others, problematic modeling assumptions regarding categorical distributions and i.i.d. sampling, and significant presentation issues affecting reproducibility. All four reviewers expressed serious concerns about these limitations, giving "reject" ratings with high confidence.

**Additional Comments On Reviewer Discussion:**

The reviewers raised detailed concerns about methodological soundness, empirical validation, and presentation clarity. While the authors provided lengthy responses attempting to address these concerns, none of the reviewers updated their scores or indicated satisfaction with the rebuttals. Most reviewers did not engage in further discussion after the authors' responses, suggesting the explanations were not sufficiently convincing to overcome the identified weaknesses. Given the unanimous reject recommendations and lack of productive discussion during the rebuttal period, this submission is being rejected.

---

### Decision · Program_Chairs · 2025-01-22

Reject